# Cultural Concept Adaptation on Multimodal Reasoning

**Zhi Li** and **Yin Zhang**[*]

College of Computer Science and Technology, Zhejiang University, China
{zhili, zhangyin98}@zju.edu.cn

| Setting | Multilingual | Multimodal | Multicultural |
|---|:---:|:---:|:---:|
| mBERT (Devlin et al., 2019) | ✓ | | |
| XLM-R(Conneau et al., 2020) | ✓ | | |
| Vilbert (Lu et al., 2019) | | ✓ | |
| Unicoder-VL (Li et al., 2020) | | ✓ | |
| M³P(*Niet al.*, 2021a) | ✓ | ✓ | |
| UC²(*Zhou et al.*, 2021) | ✓ | ✓ | |
| CCLM (Zeng et al., 2022) | ✓ | ✓ | |
| Our Method | ✓ | ✓ | ✓ |

Table 1: The different settings between our method and previous work.

## Abstract

Developing cultural adaptation methods is important, which can improve the model performance on the low-resource ones and provide more equitable opportunities for everyone to benefit from advanced technology. Past methods primarily focused on multilingual and multimodal capabilities, and the improvement of multicultural competence is still an unexplored problem. This is largely due to the difficulty of data scarcity and expensive annotation. In this paper, we navigate this uncharted territory by leveraging high-resource cultures to facilitate comprehension of low-resource ones. We first introduce an annotation-free method for cultural-concept adaptation and construct a concept mapping set. To facilitate the model's comprehension of cultural-concept mappings, we propose a new multimodal data augmentation called CultureMixup. This approach employs a three-tier code-switching strategy on textual sentences. Additionally, it uses a cultural concept-based mixup method for the images. This combination effectively generates new data instances across culture, phrase, word, and image levels. For visually grounded reasoning across languages and cultures, experimental results on five languages show that our method consistently improves performance for four existing multilingual and multimodal models on both zero-shot and few-shot settings.

## 1 Introduction

The computer vision (CV) and natural language processing (NLP) communities have witnessed significant strides in multilingual and multimodal research in recent years. For instance, XLMs (Conneau and Lample, 2019) have substantially surpassed the previous benchmarks in cross-lingual tasks. The UC² model (Zhou et al., 2021) learns to represent input tokens in a context-aware manner by leveraging both linguistic and visual content.

However, despite these advancements, the multicultural element is often neglected. The development of cultural adaptation methods is critical as they enhance model performance for low-resource languages and democratize the benefits of advanced technology. Hershcovich et al. (2022) underscores two major hurdles in cross-cultural NLP: cultural concepts and common sense. Our focus is primarily on the former: models trained on high-resource languages and images struggle to comprehend low-resource cultural concepts.

A number of previous works (Vilares and Gómez-Rodríguez, 2018; Acharya et al., 2020; Yin et al., 2021; Liu et al., 2021a; Cao et al., 2023) have delved into cultural topics, largely focusing on cultural differences or evaluating the cross-cultural competency of computational models instead of enhancing them. The primary reason is the complexity of improving cross-cultural abilities, as low-resource languages and their cultural concepts are inherently scarce, exacerbating the data scarcity issue. Moreover, annotating cross-cultural data and establishing links between concepts across cultures is an expensive process given the limited number of annotators well-versed in various countries' cultures.

To overcome this challenge, we initially propose an annotation-free method for cultural-concept adaptation, which constructs a concept mapping set. An instance of cultural-concept adaptation involves the Chinese concept Erhu [1], which has no corresponding English translation. Explaining

---

[*]Corresponding author: Yin Zhang.

it to English-speaking individuals unfamiliar with Chinese culture would likely involve likening the Erhu to a Chinese violin. This is an example of cultural adaptation. Leveraging the relationships of hypernyms, hyponyms, and synonyms from publicly accessible semantic dictionaries, our method maps source cultural concepts to their corresponding target concepts, thereby eliminating the need for costly manual annotation.

To support the model's understanding of cultural-concept mappings, we subsequently introduce a novel cultural concept-based multimodal data augmentation technique. This technique features a three-tier code-switching strategy on textual sentences and a cultural concept-based mixup method for images [2]. By training the model on both original and augmented data, we manage to significantly boost the model's performance on visually grounded reasoning tasks across languages and cultures. This improvement is reflected by a minimum increase of 2 points over existing multilingual multimodal models. Furthermore, our method can be adapted to improve specific languages or cultural topics by modifying the sampling distribution, thereby mitigating model bias.

Our contributions are encapsulated as follows:

- Leveraging web resources, we propose an annotation-free cultural adaptation method. By utilizing relationships of hypernyms, hyponyms, and synonyms from openly accessible semantic dictionaries, we construct a cultural adaptation graph that facilitates mapping between source and target cultural concepts.

- To combat data scarcity and foster the model's understanding of cultural adaptation mappings, we introduce a novel cultural concept-based multimodal data augmentation technique, generating new data instances at the concept, phrase, word, and image levels.

- Key results for the task of visually grounded reasoning across languages and cultures reveal that our methods consistently and significantly outperform baseline measures. Additionally, our technique can be tailored to enhance specific languages or cultural topics by adjusting the sampling distribution, thus reducing model bias.

---

[1] 二胡 (Erhu) is a two-stringed bowed musical instrument, which may also be called a Southern Fiddle and is sometimes known in the Western world as the Chinese violin.

## 2 Related Work

### 2.1 Cultural Research

Human language and visual content are intimately entwined with their respective cultures, evolving together, mirroring, and reciprocally influencing them. Culture is typically tied to a specific geographic region or locality, with distinct cultures characterizing different countries. The extant literature on culture tends to concentrate on three key aspects: examining cultural differences or similarities (Vilares and Gómez-Rodríguez, 2018; Acharya et al., 2020; Kozlowski et al., 2018; Sun et al., 2020), developing cross-cultural benchmarks (Peskov et al., 2021; Yin et al., 2021; Liu et al., 2021a), and evaluating the cross-cultural competence of computational models (Nguyen et al., 2022; Arora et al., 2022; Cao et al., 2023). For instance, Liu et al. (2021a) outline the multifaceted challenges involved in reasoning visually across languages and cultures, encompassing cross-modal, cross-lingual, and cross-cultural aspects. In contrast to the majority of prior research focusing predominantly on analysis and evaluation, our work confronts the issue directly by enhancing the model's adaptability to low-resource cultural concepts from both a visual and textual standpoint.

### 2.2 Code-switching and Mixup Methods

Code-switching is a common occurrence in multilingual communities, wherein the lexicon and morphemes of two or more languages are interchangeably utilized in oral or written communication. Training models using code-switched data encourages the alignment of source and target language representations by blending their contextual information. This approach has been used to challenge multilingual models (Tan and Joty, 2021), enhance Neural Machine Translation (NMT) tasks (Yang et al., 2020a; Liu et al., 2021b; Yang et al., 2020b), and further cross-lingual tasks (Qin et al., 2020; Zhang et al., 2021; Lee et al., 2021). In this study, we broaden the conventional perception of code-switching, transitioning it from a solely linguistic phenomenon to a cultural one.

While code-switching operates on sentences, mixup methods are utilized in a variety of con-

---

[2] Code-switching is a widespread phenomenon in multilingual communities, characterized by switching words and morphemes from two or more languages in speech or writing. The switched elements usually bear semantic similarity to the originals.

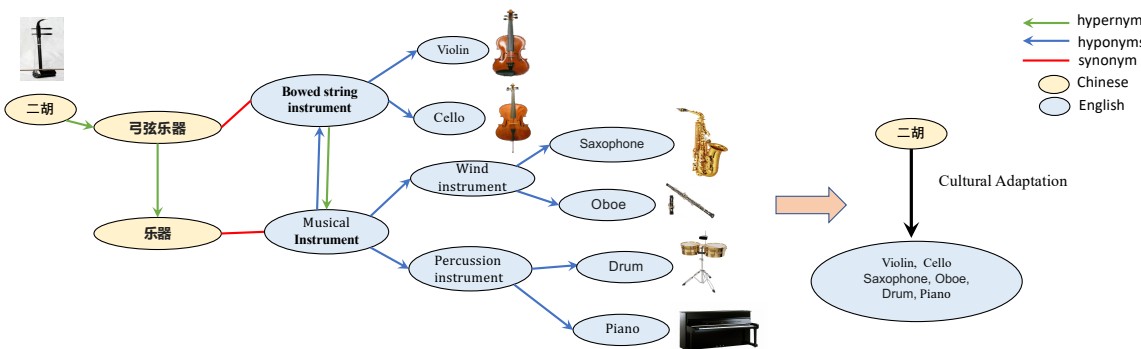

Figure 1: Consider the cultural adaptation of the term 二胡(Erhu). 二胡 translates to "Erhu" in its phonetic English form, but lacks a direct English equivalent. How might this be explained to someone living in an English-speaking country, unfamiliar with Chinese culture? You might refer to the Erhu as the "Chinese violin," as a form of cultural adaptation. We advocate a comprehensive method for cultural adaptation, utilizing relationships of hypernyms, hyponyms, and synonyms from open and freely-available semantic dictionaries to construct a cultural adaptation graph. Each leaf node within this graph could represent a potential cultural adaptation of 二胡(Erhu). The shorter the path distance in the graph, the more precise the adaptation. For instance, "violin" and "cello" are better adaptations than "saxophone" and "drum".

texts, such as mixup (Zhang et al., 2017), cutmix (Yun et al., 2019), attentive cutmix (Walawalkar et al., 2020), and alignmixup (Venkataramanan et al., 2022). Hao et al. (2023) introduced a joint data augmentation method, which generates new image-text pairs while maintaining semantic coherence through image interpolation and text concatenation. In contrast to these approaches, we substitute the target portion of the image with one that corresponds to a low-resource cultural concept.

## 2.3 Multilingual Multimodal Models

Most multimodal models based on transformer architecture (Ni et al., 2021b; Zhou et al., 2021; Jain et al., 2021; Liu et al., 2021a; Khan et al., 2021; Song et al., 2021; Zeng et al., 2022) are pre-trained using self-supervised objectives. While these methodologies advance the multilingual and multimodal capabilities of models, they often overlook cross-cultural aptitude. We propose a cultural concept adaptation approach to improve model performance across different cultures. By extending the code-switching mechanism to cultural concepts during fine-tuning, our method can help mitigate biases towards language and culture that may arise from imbalanced pre-training resource distribution, an issue that is challenging to address using self-supervised pre-training objectives alone.

## 3 Method

To overcome data scarcity and costly labeling, we initially propose an annotation-free method for

cultural-concept adaptation, which constructs a concept mapping set. To support the model's understanding of cultural-concept mappings, we subsequently introduce a novel cultural concept-based multimodal data augmentation technique. By training the model on both original and augmented data, we significantly boost the model's performance.

### 3.1 Cultural Adaptation

To illuminate the research methodology, we now formally define the central components:

**Definition 1 (Cultural Adaptation Set)**
*Consider a low-resource cultural concept $x$ and a high-resource cultural concepts set $Y = \{y_1, \ldots, y_n\}$. The cultural adaptation set of $x : Y_x = \{y_k, \ldots, y_m\}$ is a subset of $Y$ such that each $y \in Y_x$ shares similar category attributes[3] to $x$.*

**Problem Statement** Given $x \in X$, where $X$ is a set of cultural concepts, how can we identify the cultural adaptation set $Y_x$?

**Proposed Solution** We denote $H_o$, $S$, and $H_e$ as functions to query hyponyms, synonyms, and hypernyms respectively from publicly accessible semantic networks such as Conceptnet (Speer et al., 2017) and Wordnet (Miller, 1995). We construct a cultural adaptation graph $G_x$ by applying a composite function F(x), which is defined as follows:

$$F(x) = H_o(\ldots (H_o(S(H_e(\ldots (H_e(x)))))))) \quad (1)$$

---

[3]For example, the 'violin' and 'erhu' are both classified under the category of orchestral instruments.

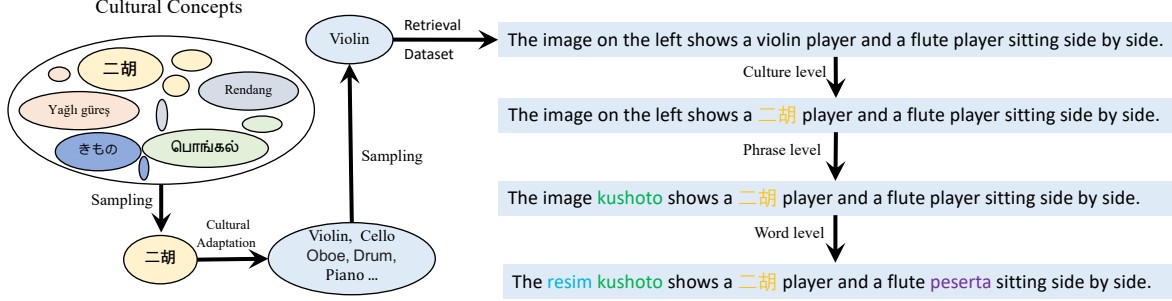

Figure 2: A schematic of three-tier code-switching strategy to the textual sentences. We commence by collecting data from web resources and forming a set of cultural concepts, each represented by an oval of a distinct color. Following this, a cultural concept is sampled for adaptation, based on a probability distribution. This selected adaptation is used to retrieve a corresponding data instance from the training dataset. Subsequently, we apply three levels of code-switching to the retrieved sentences. In these code-switched sentences, words highlighted in yellow, green, blue, and purple denote languages — Chinese, Swahili, Turkish, and Indonesian, respectively.

**Definition 2 (Cultural Adaptation Graph)**

*A cultural adaptation graph $G_x = (V, E)$, where $V$ denotes the set of vertices and $E$ the set of edges, is a directed graph created by the iterative application of the Hypon, Syn, and Hyper functions starting from the cultural concept $x$. Each vertex $v_i \in V$ represents a cultural concept linked to $x$ through a path of hyponym, synonym, or hypernym relationships, while each directed edge $e_{ij} \in E$ depicts one of these relationships $\{H_o, S, H_e\}$ between the cultural concepts $v_i$ and $v_j$.*

Given a cultural adaptation graph $G_x = (V, E)$, each leaf node $v_i$ in $G_x$ potentially represents a cultural adaptation of the cultural concept $x$. An example of this process is illustrated in Figure 1, showing the cultural adaptation of "二胡(Erhu)". The 'Erhu' has a hypernym "弓弦乐器 [4]", synonymous in English with "Bowed string instrument". This term has various hyponyms, including the 'violin' and 'cello', making them potential cultural adaptations of the 'Erhu'. Similarly, "乐器", the hypernym of "弓弦乐器", and by extension a second-order hypernym of 'Erhu', translates to "Musical Instrument" in English. Thus, other instruments like the 'saxophone' and 'oboe' may also serve as potential cultural adaptations of the 'Erhu'. In fact, every leaf node in the cultural adaptation graph could potentially represent a cultural adaptation of 'Erhu', where shorter path distances indicate higher accuracy. For instance, 'violin' and 'cello' would provide more accurate cultural adaptations than 'saxophone' and 'drum'. A simple iterative traversal algorithm can yield all leaf nodes and their respective path distances from 'Erhu'. These

cultural adaptation sets can serve as a kind of multi-cultural resource.

## 3.2 Culture Mixup

To mitigate the issue of data scarcity and prompt the model to comprehend the mapping of cultural adaptation, we propose a cultural concept-based multimodal data augmentation named CultureMixup. It includes a three-tier code-switching strategy on the textual sentences and a cultural concept-based mixup method on the image and thereby generates new data instances from the culture, phrase, word, and image levels. We initially gather data from online resources to construct a set $X$ of cultural concepts, each paired with corresponding images. Refer to the appendix A for further details. Subsequently, we select a cultural concept $x \in X$ for cultural adaptation according to a predefined probability distribution. The sampling of an adaptation $y_k \in Y_x$ from the high-resource cultural adaptation set is inversely proportional to its path distance in the cultural adaptation graph $G_x$ (as defined in Equation 1), i.e., $P(y_k|x) \propto \frac{1}{d(x,y_k)}$. The adapted concept $y_k$ is then used to retrieve data instances from the training dataset. Each data instance comprises both textual sentences and images.

### 3.2.1 Three-tier Code-switching on Text

We apply a three-tier code-switching strategy to the textual sentences, as displayed in Figure 2.

---

[4] 弓弦乐器 is bowed string instruments in English, which are a subcategory of string instruments that are played by a bow rubbing the strings. The bow rubbing the string causes vibration which the instrument emits as sound.

**Cultural-level**   We extend the conventional understanding of code-switching from a purely linguistic phenomenon to a cultural one. The culturally adapted concept $y_k$ in the sentence is replaced by the original low-resource cultural concept $x$. For instance, in Figure 2, the word "violin" is code-switched with its Chinese cultural adaptation "二胡".

**Phrase-level**   For the phrase-level code-switching, we collect and identify high-frequency phrases pertinent to reasoning and translate them into different languages. If a sentence contains these selected phrases, they are code-switched based on a given probability. For example, in Figure 2, the phrase "on the left" is code-switched by "kushoto" in Swahili.

**Word-level**   At the word level, each word in a sentence undergoes code-switching in a different language based on a specified probability. For example, in Figure 2, the words "image" and "player" are code-switched by "resim" in Turkish and "peserta" in Indonesian, respectively.

### 3.2.2   Concept-Based Mixup for Images

For the image-level cultural adaptation, we first apply an object detection algorithm to the images. We identify visual elements that correspond to high-resource cultural concepts $y_k$, and replace them with visual elements associated with the original low-resource cultural concept $x$. The specific process is as follows:

1. **Visual Element Detection:** We use the high-resource concept as the vocabulary term and deploy the object detection model to locate the corresponding visual elements in the image.

2. **Bounding Box Extraction:** Based on the object detection results, we extract the object that needs to be replaced from the original high-resource image using bounding boxes.

3. **Resizing:** We adjust the size of the low-resource image element, which is collected together with concepts themselves, to match the size of the bounding box.

4. **Pasting:** The resized new low-resource image is then pasted into the corresponding location of the original image.

5. **Smoothing and Harmonization:** Finally, smoothing and harmonization techniques are

applied to seamlessly integrate the new object into the original image and make it more compatible with the background.

This introduces cultural diversity at the visual level, thereby enriching the data instances for training.

### 3.3   Reduce Model Bias

Pretrained multilingual multimodal models often demonstrate disparate performance across various languages and cultural contexts in test datasets, a discrepancy likely attributable to uneven resource distribution during pretraining. These imbalances pose challenges that are not readily addressed by self-supervised pretraining objectives. Our method offers a mechanism to ameliorate these language and cultural biases by manipulating the sampling distribution. Essentially, we can enhance model performance on specific language or cultural topics in a controllable manner. For instance, if the model is anticipated to be applied in a Turkish context, the sampling probability for Turkish can be increased during data augmentation. Likewise, if the model will be deployed in scenarios involving traditional Chinese musical instruments, like the Erhu, we can elevate the sampling probability of these specific Chinese musical concepts. In summary, our approach provides a statistically significant, fine-grained performance boost for the model over predefined language or cultural categories.

## 4   Experiment

### 4.1   Dataset

We collect and evaluate cultural concepts from diverse cultures (Indonesian, Swahili, Tamil, Turkish, Chinese) using crowd-sourced workers. The process involves gathering a wide range of cultural concepts manually and validating them through a majority vote. For a detailed methodology, please refer to the appendix. This approach, while cost-effective, emphasizes accuracy by requiring a significant consensus among evaluators and prioritizing manual collection to capture cultural nuances. This dataset is publicly available [5]. In evaluating the resulting cultural adaptation graphs, about 84% aligned with human judgment, confirming the method's effectiveness. For the less accurate graphs, issues were primarily due to translation limitations for certain cultural concepts' hypernyms.

---

[5]https://github.com/zhilizju/Culture-mixup

| | | NLVR2 | MaRVL | | | | | |
|---|---|---|---|---|---|---|---|---|
| Model | $x:y$ | EN | ZH | TA | SW | ID | TR | avg. |
| | | | *Proportion* | | | | | |
| *mUNITER* | 1:1 | 69.23 | 54.85 | 55.44 | 53.01 | 53.43 | 57.2 | 54.79 |
| | 2:1 | 69.74 | 56.99 | 55.81 | 53.26 | 54.45 | 56.67 | 55.44 |
| | 3:1 | 70.45 | 57.33 | 56.02 | 53.97 | 55.14 | 58.67 | 56.23 |
| | 4:1 | 70.82 | 58.04 | 55.57 | 53.61 | 54.92 | 57.99 | 56.03 |
| | 5:1 | 71.24 | 56.88 | 55.41 | 52.99 | 53.24 | 56.98 | 55.10 |
| *mUNITER* | 3:1 | 70.45 | 57.33 | 56.02 | 53.97 | 55.14 | 58.67 | 56.23 |
| | 9:3 | 71.22 | 58.47 | 56.11 | 54.19 | 55.24 | 58.91 | 56.58 |
| | 15:5 | 71.56 | 59.71 | 58.45 | 54.35 | 55.24 | 58.81 | 57.31 |

Table 2: Different proportion on model performance. $x:y$ means that the total dataset is $x+y$ times larger than the origin English training data. And $\frac{x}{x+y}$ of the total dataset is the origin English training data (just copy origin dataset x times) and $\frac{y}{x+y}$ of the total dataset is augmented data. The results cover five languages: Indonesian (ID), Swahili (SW), Tamil (TA), Turkish (TR) and Mandarin Chinese (ZH).

*Zero-Shot*

| Model | Method | NLVR2 EN | MaRVL ZH | TA | SW | ID | TR | avg. | *Translate-Test* |
|---|---|---|---|---|---|---|---|---|---|
| *mUNITER* | MaRVL(Liu et al., 2021a) | 73.2 | 56.8 | 52.2 | 51.5 | 55.0 | 54.7 | 54.0 | 63.5 |
| | IGLUE (Bugliarello et al., 2022) | 71.91 | 55.34 | 52.66 | 51.17 | 54.79 | 54.66 | 53.72 | 63.82 |
| | Our Reproduce | 70.93 | 54.05 | 53.14 | 51.72 | 55.41 | 53.22 | 53.51 | - |
| | + Our Method | 71.56 | **60.44 (59.71)** | **58.92(58.45)** | **54.87(54.35)** | **55.72(55.24)** | **59.53(58.81)** | **57.90(57.31)** | - |
| *xUNITER* | MaRVL(Liu et al., 2021a) | 72.8 | 55.0 | 55.1 | 55.5 | 57.1 | 58.0 | 56.1 | 64.4 |
| | IGLUE (Bugliarello et al., 2022) | 71.55 | 53.06 | 53.06 | 55.51 | 55.14 | 56.19 | 54.59 | 64.04 |
| | Our Reproduce | 70.81 | 54.35 | 53.95 | 55.33 | 54.79 | 58.31 | 55.34 | - |
| | + Our Method | 70.34 | **57.91 (56.78)** | **58.03(57.88)** | **60.01(59.06)** | **58.78(58.21)** | **59.67(59.44)** | **58.88(58.27)** | - |
| $UC^2$ | IGLUE (Bugliarello et al., 2022) | 70.56 | 59.88 | 60.47 | 52.62 | 56.74 | 56.70 | 57.28 | 63.09 |
| | Our Reproduce | 69.05 | 58.3 | 60.06 | 52.97 | 57.18 | 56.102 | 57.07 | - |
| | + Our Method | 68.88 | **60.13 (59.73)** | **61.61(60.88)** | **58.55(57.97)** | **58.91(58.75)** | **60.72(60.34)** | **59.98(59.33)** | - |
| $M^3P$ | IGLUE (Bugliarello et al., 2022) | 68.22 | 55.04 | 56.04 | 55.69 | 56.47 | 56.78 | 56.00 | 62.52 |
| | Our Reproduce | 68.71 | 52.96 | 57.73 | 53.70 | 58.16 | 58.22 | 56.15 | - |
| | + Our Method | 68.10 | **59.21(58.73)** | **60.89(59.96)** | **61.23(60.6)** | **59.18(58.99)** | **61.31(60.65)** | **60.36(59.79)** | - |

Table 3: Zero-shot statistics. The best scores are in bold. For our method, we not only report the best performance but also report the statistical mean. The digits in "( )" represent the average value of different random seeds. The results cover five languages: Indonesian (ID), Swahili (SW), Tamil (TA), Turkish (TR) and Mandarin Chinese (ZH).

Strategies like restricting the use of higher-order hypernyms and implementing an exponential decay of sampling probability for concept inclusion were employed to enhance accuracy and authenticity, ensuring the graphs' overall quality and reliability. See the appendix B for details.

We use the Detic model (Zhou et al., 2022) available at Facebook Research's GitHub repository for object detection. It employs the CLIP (Radford et al., 2021) classifier and offers open-vocabulary capabilities. With this model, we can freely configure the vocabulary, allowing us to detect specific cultural concepts within images. If you want a more precise result, it is recommended to use the segment anything model(Kirillov et al., 2023), but it may require some manual clicking.

Following Liu et al. (2021a), we employ NLVR2 in English for training and MaRVL for testing. MaRVL (Liu et al., 2021a) is a visually grounded reasoning dataset similar to NLVR2 (Suhr et al.,

2019). It not only spans five typologically diverse languages (Chinese, Tamil, Swahili, Indonesian, and Turkish) but also adopts different basic-level concepts across different cultures. Thus, the challenges are multi-faceted, including cross-modal, cross-lingual, and cross-cultural aspects. In this task, given two images ($I_{left}$ and $I_{right}$) and a description **D**, the model needs to assess the validity of the description given the images, which can be cast as a classification problem. See the appendix C for sampled examples and detailed descriptions.

### 4.2 Baseline

We assess the efficacy of our approach using four existing multilingual multimodal pretrained models that have been released so far:

- *mUNITER* (Liu et al., 2021a), which extends the UNITER architecture (Chen et al., 2020) to support multilingual functionality.

| | | Shot | | | | |
|---|---|---|---|---|---|---|
| | | 1 | 2 | 4 | 10 | 10x2 |
| $mUNITER$ | ZH | 59.41/56.52 | 59.23/54.84 | 60.18/54.55 | 60.60/55.34 | 61.63/54.05 |
| | TR | 59.66/52.54 | 59.24/55.93 | 59.73/55.59 | 60.48/54.66 | 60.97/54.75 |
| | ID | 55.50/51.42 | 55.88/52.13 | 56.34/54/34 | 56.71/56.83 | 56.97/51.42 |
| $xUNITER$ | ZH | 57.02/54.45 | 56.72/53.66 | 57.63/54.94 | 57.66/53.46 | 57.82/55.73 |
| | TR | 59.19/55.93 | 59.58/57.46 | 59.48/57.54 | 59.86/57.80 | 60.60/58.05 |
| | ID | 58.60/56.12 | 59.48/57.18 | 59.55/58.87 | 60.51/58.87 | 60.75/58.60 |
| $UC^2$ | ZH | 59.34/58.99 | 59.40/57.02 | 59.29/58.99 | 59.99/57.21 | 60.91/60.18 |
| | TR | 59.64/55.76 | 59.92/54.39 | 60.71/52.63 | 61.15/55.93 | 61.53/56.27 |
| | ID | 59.33/56.12 | 59.49/56.29 | 59.52/57.53 | 60.11/57.62 | 60.89/58.51 |
| $M^3P$ | ZH | 58.92/52.47 | 59.32/53.85 | 59.71/52.47 | 60.03/54.45 | 60.12/54.64 |
| | TR | 59.90/57.88 | 59.58/58.98 | 59.81/57.97 | 60.59/58.39 | 61.19/57.54 |
| | ID | 58.69/57.36 | 59.40/56.92 | 59.61/57.27 | 59.57/58.51 | 59.91/58.51 |

Table 4: Different shots statistics. Results of Our methods/baseline. The baseline results come from IGLUE (Bugliarello et al., 2022). The results cover three languages: Indonesian (ID), Turkish (TR), and Mandarin Chinese (ZH).

- $xUNITER$ (Liu et al., 2021a), which is identical to $mUNITER$, except for its initialization: while $mUNITER$ is initialized from mBERT, xUNITER originates from XLM-R.

- $M^3P$ (Ni et al., 2021b), which introduces pretraining tasks using multimodal code-switching.

- $UC^2$ (Zhou et al., 2021), which uses masked region-to-token modeling and visual translation language modeling as pretraining tasks.

To provide a fair comparison with baselines (Liu et al., 2021a; Bugliarello et al., 2022), we adopt nearly identical experimental setups and hyperparameters except that we finetune models on the origin and augmented NLVR2 (Suhr et al., 2019) dataset. Despite augmenting the dataset, we maintain the same total number of training steps by reducing the training epochs. For more detailed information about settings and the implementation of the model, please refer to the appendix D. Our code is based on VOLTA (Bugliarello et al., 2020).

## 4.3 Results

In this part, we mainly discuss four parts with experimental results. (1) What proportion is appropriate for the augmented data? (2) The zero-shot and few-shot performance of models with our proposed methods.(3) Ablation studies. (4) Controllability and reduce model bias.

### 4.3.1 Proportion

We conduct two groups of experiments on $mUNITER$ in a zero-shot setting: one examines the impact of the proportion of augmented data, while the other investigates the effect of the total volume of augmented data on model performance while keeping the proportion constant. In Table 2, $x : y$ implies that the total dataset is $x + y$ times larger than the original English training data. Furthermore, $\frac{x}{x+y}$ of the total dataset is the original English training data (essentially replicating the original dataset $x$ times), and $\frac{y}{x+y}$ is the code-switched generated data. To maintain the total number of training steps, we reduce the training epochs and train the models for $\frac{20}{x+y}$ epochs.

For the first group of experiments, we establish five different ratios $x : y$=1 : 1, 2 : 1, 3 : 1, 4 : 1, 5 : 1. Results in Table 2 suggest that as the volume of original data increases, the model's performance on the English test set consistently improves. However, performance in other languages and cultures initially increases then decreases. This reveals a trade-off: while a substantial proportion of English data aids the model's task comprehension, an appropriate amount of augmented data from other cultures facilitates the model's transfer ability. A ratio of roughly 3:1 yields optimal results. We further investigate this by holding the scale constant and incrementally increasing the English and augmented datasets.

In order to examine the influence of the total volume of augmented data on model performance

| | | Ablation | | | | | | |
|---|---|---|---|---|---|---|---|---|
| Model | Method | NLVR2 | MaRVL | | | | | |
| | | EN | ZH | TA | SW | ID | TR | avg. |
| $mUNITER$ | Baseline | 70.93 | 54.05 | 53.14 | 51.72 | 55.41 | 53.22 | 53.51 |
| | Our Method | 71.56 | 59.71 | 58.45 | 54.35 | 55.24 | 58.81 | 57.31 |
| | All w/o Concept | 71.18 | 58.87 | 57.3 | 53.82 | 54.61 | 57.52 | 56.42 |
| | All w/o Phrase | 70.71 | 57.95 | 56.94 | 53.18 | 54.02 | 57.19 | 55.86 |
| | All w/o Word | 70.65 | 58.69 | 57.22 | 53.55 | 54.38 | 57.55 | 56.28 |
| | All w/o Image | 70.85 | 58.14 | 56.81 | 52.95 | 54.69 | 56.33 | 55.78 |

Table 5: Ablation studies across five languages.

while keeping the proportion constant, we establish three different ratios $x : y$=3 : 1, 9 : 3, and 15 : 5. As indicated in Table 2, the performance of the model improves with an increase in the volume of augmented data. Taking into account the results from these two sets of experiments, we decide to amplify the dataset to twenty times the original size and choose a ratio of $x : y = 15 : 5$ for our subsequent zero-shot and few-shot experiments. Although we do not contend that the ratio of $x : y = 15 : 5$ is optimal, we assert that this choice is adequate to demonstrate the effectiveness of our approach.

### 4.3.2 Zero-shot and Few-shot

As previously mentioned, we employ a configuration of $x : y = 15 : 5$ to augment the dataset to twenty times its original size. This involves replicating the original English NLVR2 dataset (Suhr et al., 2019) 15 times and generating a augmented dataset five times larger. Consequently, the final dataset is 20 times the size of the original English NLVR2 dataset. To maintain a consistent number of training steps with Liu et al. (2021a); Bugliarello et al. (2022), who trained models for 20 epochs, we train our models on this expanded dataset for a single epoch. We present both the best results and statistical mean for our method. 'Translate-test' refers to the setup in which the multilingual MaRVL datasets are translated into English. The results of Liu et al. (2021a); Bugliarello et al. (2022) are used directly as zero-shot benchmarks.

Table 3 displays the zero-shot performance of the four models and demonstrates that our method consistently and statistically surpasses the baselines. Our approach considerably diminishes the disparity between the performance in the translation test and the transfer performance, validating the effectiveness of our code-switching method. We observe that, compared with the baselines, our

method enhances the $M^3P$ and $mUNITER$ scores by about $3 \sim 4$ points, while $UC^2$ and $xUNITER$ gain only about $2 \sim 3$ points. This disparity might stem from the fact that $UC^2$ and $xUNITER$ have acquired better-aligned representations during the pre-training phase.

Table 4 displays the results of few-shot performance on three languages, demonstrating that our method also achieves the state-of-the-art performance in the few-shot setting. Nevertheless, similar to Liu et al. (2021a); Bugliarello et al. (2022), our results corroborate the finding that unlike text-only multilingual tasks, where even a handful of examples in the target language can substantially enhance model performance, this phenomenon is largely absent in multimodal multilingual settings (Bugliarello et al., 2022). As the number of shots increases, the model's performance remains largely unchanged or shows slight growth. We attribute this to two main factors. Firstly, the inherent complexity of the task, and secondly, within the same language, data samples embodying diverse cultural concepts may vary significantly. The model may overfit to data samples associated with specific cultural concepts, a phenomenon that warrants further investigation in future studies.

### 4.3.3 Ablation Study

To examine the impact of our cultural concept-based multimodal data augmentation approach, we conducted an ablation study on $mUNITER$, maintaining the 15:5 setting for consistency. The results, as presented in Table 5, represent the statistical mean of various random seeds and underscore that each component of our method significantly contributes to enhancing the model's performance.

### 4.3.4 Reduce Model Bias

Our method can be utilized to mitigate model bias on specific target languages or cultural topics by

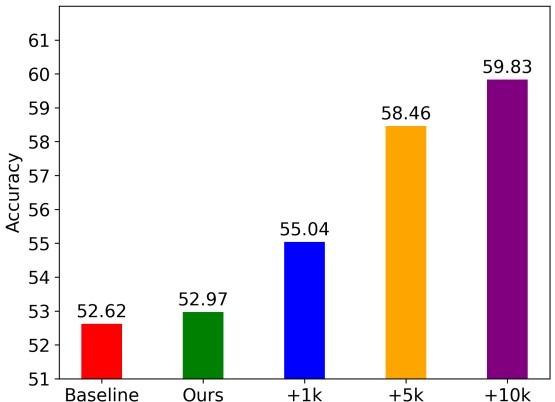

Figure 3: $UC^2$ on Swahili language. "Baseline" represents the score reported by Bugliarello et al. (2022) and "Ours" represents our replication of basline. "+1k" /"+10k" represent adding 1k/10k Swahili-English code-switched data to training dataset. "Best" represent the best performance when we add more code-switched data examples.

adjusting the sampling distribution. For instance, most models exhibit relatively lower scores in Swahili compared to other languages, with $UC^2$ being particularly affected. To address this, we specifically focus on improving $UC^2$'s performance in Swahili. By sampling, we obtain 1k and 10k examples of Swahili-English code-switched data, which we then merge with the original English dataset to create a new training dataset. As Figure 3 illustrates, our method can significantly enhance the performance of $UC^2$ in Swahili. We also enhance the model's performance on the topic "Speech and language" in Chinese. For further details, please refer to appendix D.

## 5 Conclusion

To attack the difficulties of data annotation and scarcity, we propose an annotation-free cultural adaptation method and design a novel cultural concept-based multi-modal data augmentation to generate the new data example. By training the model on the augmented dataset, key results indicate that our methods consistently and statistically outperform the baselines. In the future, we plan to apply our method to more downstream tasks related to culture. Employing curriculum learning and designing more refined training strategies according to the difficulty of different languages and cultural concepts is also worth exploring. At the same time, how to further extend our method to make it more applicable to multi-modal models based on autore-

gressive generation, such as GPT-4-V [6], is also highly worthwhile to explore.

## Limitations

Our approach principally amplifies the conceptual adaptability of models to low-resource cultures. Nevertheless, cultural differences are complex and multidimensional, encompassing not only conceptual elements but also elements of common sense. The comprehensive acquisition of such common sense across diverse cultures is a vital yet challenging endeavor. Therefore, our community still has a considerable path to tread in order to fully enhance the multicultural competencies of AI models. Simultaneously, we only conducted experiments on multi-modal models based on masked language modeling. Further investigation is needed to determine the effectiveness on multi-modal models based on autoregressive generation.

## Acknowledgments

This work was supported by Zhejiang Provincial Natural Science Foundation of China under Grant No. LZ23F020009, the NSFC project (No. 62072399), MoE Engineering Research Center of Digital Library, China Research Centre on Data and Knowledge for Engineering Sciences and Technology, and the Fundamental Research Funds for the Central Universities. We would like to express our sincere gratitude to the anonymous reviewers for their invaluable feedback and constructive comments, which significantly contributed to the improvement of this paper.

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

## A   Collecting the Cultural Concepts

First, we need to collect the cultural concepts of different countries. In detail, we choose a diverse set of cultures and languages following: Indonesian, Swahili, Tamil, Turkish, Chinese. Each culture contains about ten chapters: festival, music, religion and belief, animal and plant, food, clothing, building, agriculture, tool, and sport. Each chapter contains several to dozens of cultural concepts. The collection of cultural concepts is primarily carried out manually. Importantly, this approach doesn't necessitate cross-cultural specialists; rather, it employs crowd-sourced workers familiar with the respective culture, making it a relatively cost-effective and simple process. The procedure is twofold:

1. **Diverse Candidate Collection**: For each culture under consideration, we involve five crowd-sourced workers and require them to explore a minimum of three types of online resources. (1) Wikipedia. For example, the article "Culture of Turkey" on Wikipedia lists many cultural concepts including foods, festivals, architecture, and so on. (2) Official websites. Most countries provide official websites to introduce their culture. (3) Search engines. Some websites retrieved by search engines such as travel guides will introduce the local culture. They collect as many cultural concepts as possible for each category. The collected data from each worker is then aggregated.

2. **Voting for Filtering**: Next, an additional 10 crowd-sourced workers from the respective country or region assess whether each gathered concept genuinely belongs to the local culture. If seven or more evaluators agree, the concept is added to the final 'culture concept set.'

This method ensures quality in several ways:

- **Inclusion of Multiple Evaluators**: By involving multiple people from the respective culture in both the collection and the validation process, we minimize individual bias and enhance the dataset's reliability.

- **Threshold for Inclusion**: The use of a voting system provides a safety net against inaccuracies and biases. If a concept is included, it's

because a significant majority (at least 7 out of 10) of the evaluators from that culture have vouched for its relevance.

- **Manual Over Automatic**: While automated methods may miss nuances or make errors, manual collection engages those who understand the cultural intricacies best—the people from that culture.

Hence, our approach offers a robust, yet economical way of collecting high-quality cultural concepts.

## B   Human Evaluation of Cultural Adaptation Graphs

The evaluation of the generated cultural adaptation graphs yielded quite reassuring results:

- **High Human Agreement:** Approximately 84% of the generated cultural adaptation graphs were in alignment with human judgment. This high rate of agreement underscores the validity of our approach in capturing culturally meaningful concepts.

- **Addressing Inaccuracies:** For the remaining graphs that were less accurate, the primary issue often lay in the lack of direct English translations for the low-resource cultural concepts' hypernyms. To address this, the method would need to consider higher-order hypernyms, making the entire path too distant and potentially distorting cultural similarity according to human evaluators.

To counteract these issues, we implemented the following strategies:

- **Hypernym Limitation:** As described in the paper, we restrict the method to considering at most three-order hypernyms during the construction of the cultural adaptation graph. If a suitable translation or hypernym cannot be identified within this constraint, the concept is discarded.

- **Exponential Decay of Sampling Probability:** The paper also outlines that the sampling probability for including concepts in the cultural adaptation set decays exponentially with path distance. This mechanism serves to mitigate possible inaccuracies by giving greater weight to more closely related concepts.

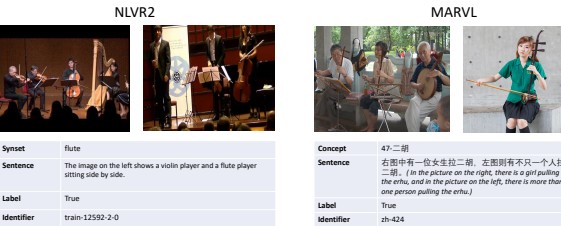

Figure 4: Sampled examples from NLVR2 (Suhr et al., 2019) and MaRVL(Liu et al., 2021a). The NLVR2 is a semantically-diverse dataset for reasoning about natural language descriptions of photos. The task is to determine if a caption is true with regard to a pair of images. MaRVL is similar to NLVR2, except that the descriptions of NLVR2 are in English and MaRVL spans 5 typologically diverse languages (Chinese, Tamil, Swahili, Indonesian, and Turkish).

These measures are designed to ensure that the cultural adaptation graph generated is of high quality, both in terms of capturing authentic cultural elements and in conforming to human judgment. Therefore, while the graph is not perfect, it is constructed with numerous safeguards to ensure its utility and accuracy.

## C   NLVR2 and MaRVL Dataset

Liu et al. (2021a) points that most of the synsets employed by NLVR2 (Suhr et al., 2019) and ImageNet (Deng et al., 2009) are only present in 30 or fewer languages and they contain overly specific concepts that belong to leaf nodes in WordNet. Given the biases in ImageNet-derived or inspired datasets, they define a protocol to collect data that is driven by native speakers of a language, consisting of concepts arising from their lived experiences. As a consequence, the descriptions are written in five languages: Indonesian, Mandarin Chinese, Swahili, Tamil, and Turkish, and the concepts are selected to be culturally relevant. Both multilingual and monolingual models perform comparably well in English (NLVR2). When these models are evaluated on the languages in MaRVL, however, the performance of zero-shot multilingual baselines dramatically drops a lot, floating just above the chance level. Further analysis shows that there are two sources of difficulty that make MaRVL challenging: 1) cross-cultural transfer (out-of-distribution concepts with respect to English datasets) and 2) cross-lingual transfer.

## D  The Implementation in Details

### D.1  Experimental Setting

We provide a detailed experimental setting. First, we collate and merge data from three web resources and build an initial dataset containing cultural concepts of different countries. Refer to appendix A for detail. For cultural adaptation, "$/r/IsA$" in Conceptnet can be used to query hypernyms in target languages, and "$/r/Synonym$" is employed to query synonyms in English. And we use the $hyponyms()$ function of Wordnet to query hyponyms of English concepts. We only consider three-order hypernym at most. And the sampling probability decays exponentially with path distance. For phrase level, we count n-gram phrases. $n$ range from 2 to 5. We select phrases related to reasoning and translate them to other languages via Google Translate. For word level, we employ the "$/r/Synonym$" in Conceptnet.

### D.2  Model Implementation

Most multilingual multimodal pre-trained models architecture consists of a stack of Transformer layers are similar to BERT. Their inputs are the concatenation of language and vision embeddings. The language inputs are tokenized and surrounded by two special tokens $\{[CLS], t_1, ..., t_n, [SEP]\}$. The language embeddings are then obtained by index token ids just like the original BERT. The vision input consists of a set of visual features produced by a well-trained object detector. Following (Liu et al., 2021a) and (Bugliarello et al., 2022), we also add a special feature [IMG] that encodes the entire image, $\{[IMG], v_1, ..., v_m\}$. Each feature is embedded using a BERT-like embedding layer by using its bounding box coordinates as the input position. Then these language and vision embeddings will be fed into a BERT-like encoder and get $h^l_{[IMG]}, h^l_{[CLS]}, h^r_{[IMG]}, h^r_{[CLS]}$ representations at the last layer. We follow Liu et al. (2021a) and employ the cross-entropy loss function. We apply a two-layer MLP with a GeLU activation function (Hendrycks and Gimpel, 2016) on top of the image–text representation. The probability that they are both correct is predicted by a softmax over two classes (representing true and false labels):

$$P(C \mid I_l, I_r, D) = \text{softmax}\left(\text{MLP}\left(\begin{bmatrix} h^l_{[IMG]} \odot h^l_{[CLS]} \\ h^r_{[IMG]} \odot h^r_{[CLS]} \end{bmatrix}\right)\right) \tag{2}$$

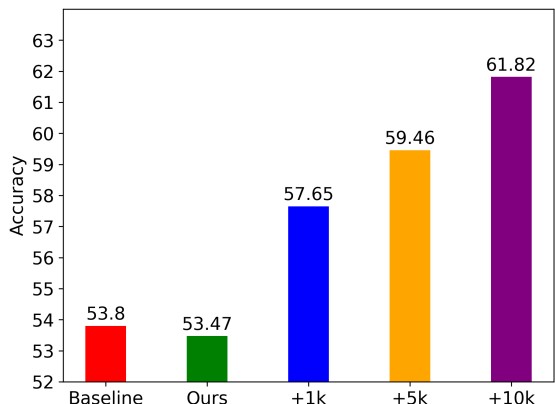

Figure 5: $mUNITER$ on the topic *Speech and language* of Chinese test dataset in MARVL. "MaRVL" represent the score reported by Liu et al. (2021a) and we also reproduce this result. "+1k" /"+10k" represent adding 1k/10k code-switched data about traditional Chinese musical instruments to the training dataset. "Best" represent the best performance when we add more code-switched data examples.

## E  Error Case Analysis

### E.1  Performance Across Languages

As demonstrated in Table 3 of the paper, performance diverges significantly across languages. This discrepancy is due largely to the imbalanced distribution of pre-training data, linguistic idiosyncrasies, and cultural distinctions. Furthermore, the ease of language transfer from English to other languages varies considerably.

### E.2  Within-Language Category Differences

An intra-language examination reveals that performance fluctuates across different categories. For instance, the model demonstrates heightened efficacy in the "animal and plant" and "building" categories relative to others. Section 4.3.4 elaborates on this phenomenon, attributing it primarily to the distribution of pre-training data.

### E.3  Instance-Level Analysis

We sampled a subset of the test data, and when we conducted a detailed error investigation at the individual instance level, we discovered several illuminating patterns:

- **Poor Spatial Awareness:** The model predominantly falters in spatial reasoning tasks. Notably, descriptions formatted as "one image..., another image..." are more challenging for the model than those specifying "left image...,

right image...", with performance improving statistically across languages in the latter case (58.4% vs 53.8%).

- **Limited Textual Reasoning:** Performance enhances when the description is condensed into a single sentence ("in both images...") rather than dispersed across two ("one image..., another image..."), with a performance leap to 60.4%.

- **False Bias:** The models exhibit a pronounced propensity to predict 'false' in the test set, constituting 57% of the model's predictions, contrasted with 43% of 'true' predictions. This imbalance, especially conspicuous given the actual 50-50 distribution of 'true' and 'false' labels in the training and test sets, stems predominantly from the model's unfamiliarity with certain cultural concepts and images, prompting a 'false' prediction.

These analyses collectively provide a comprehensive insight into the model's limitations, illuminating potential avenues for refinement.

## F Reduce Model Bias

According to Liu et al. (2021a), each language test dataset can be divided into multiple topics based on the categories of cultural concepts. For example, the topic "Speech and language" of the Chinese test dataset in MARVL contains many traditional Chinese musical instruments, such as Erhu. Similar to adjusting the sampling probability of language, we can generate Chinese-English code-switched by adding the sampling probability of assigned cultural concepts and phrases of Chinese. Assuming models need to improve their reasoning ability on datasets about traditional Chinese musical instruments, all we need is to improve the sampling probability of concepts about this topic in our collected cultural concepts set. Figure 5 shows that as we add more related data, the model performance is becoming better on corresponding topics.

## G LLM for Cultural Concept Adaptation

This section discusses the potential of using large language models (LLMs) like ChatGPT for cultural concept adaptation. Despite their proficiency in explaining various cultural concepts, several factors discourage their exclusive use as a baseline or expert system.

- **Language Resource Limitations:** Prior studies (Jiao et al., 2023; Vilar et al., 2022; Cao et al., 2023; Havaldar et al., 2023) and our case studies on GPT-3.5's initial version indicate subpar performance in certain low-resource languages, such as Swahili (sw), despite proficiency in languages like Chinese (zh).

- **Cost-Effectiveness:** Dictionary lookups are generally more efficient and economical than operating large language models.

- **Reproducibility & Hallucinations:** Inherent weaknesses in LLMs include instability based on prompts and difficulty ensuring reproducibility. Verifying outputs is particularly challenging when hallucinations occur.

- **Dataset Compatibility:** Our downstream dataset, NLVR2, revolves around a WordNet subset, facilitating queries. Relying on Chat-GPT would necessitate additional output processing.

- **Leveraging Existing Dictionary Resources:** Our method can fully exploit existing dictionary resources for extremely low-resource languages, like Navajo, through platforms such as ConceptNet.

### G.1 Evaluating Recent Models

Recently, we evaluated the latest GPT-3.5 and GPT-4 models for cultural adaptation capabilities. Using a chain of prompt approach to construct a cultural adaptation graph yielded promising results, particularly with GPT-4. Nonetheless, these models also encounter challenges, such as potential hallucinations, and necessitate meticulous prompt design for diverse cultural adaptations. We believe that the integration of dictionary-based cultural adaptation methods with large language models represents a promising hybrid approach. For instance, a dictionary-based method could produce a broad set of cultural adaptations, subsequently filtered for relevance by a large language model. Incorporating LLMs to devise our cultural adaptation graph is indeed promising and under consideration for future exploration.