# OpenReview forum: "Cultural Concept Adaptation on Multimodal Reasoning"
_EMNLP/2023/Conference — EMNLP 2023 Main_

### Official Review · Reviewer_bqm3 · 2023-07-20

**Soundness:** 3

**Excitement:**

3: Ambivalent: It has merits (e.g., it reports state-of-the-art results, the idea is nice), but there are key weaknesses (e.g., it describes incremental work), and it can significantly benefit from another round of revision. However, I won't object to accepting it if my co-reviewers champion it.

**Paper Topic And Main Contributions:**

This work proposes a data augmentation approach for multilingual multimodal reasoning. Specifically, it constructs an external cross-language concept mapping graph to associate the concepts between different cultures (language), and calculates the correlation between concepts of different cultures through the graph path distance. Then, based on the correlation between cross-language concepts, the concepts of original dataset are replaced at word-level and phrase-level for training; image objects are also considered as replacement objects. Experimental results show that the proposed method achieves significantly better performance than the baselines.

**Questions For The Authors:**

1. Compared with the baseline method mUNITER, the performance degradation of all ablation models was not obvious. Which data augmentation strategy played a role in performance improvement? Might it be possible to ablate multiple data augmentation strategies simultaneously to observe changes in model performance?

2. I did not find relevant details about the Concept-Based Mixup method in the paper. What object detection algorithm do you use to identify images? How do you replace identified high-resource image elements with low-resource cultural image elements?

**Reasons To Accept:**

1. The paper is well-written and easy to understand.

2. This paper builds an external cross-language concept mapping graph, replacing the words, phrases and concepts in the dataset based on the graph path distance, so as to expand the dataset for training. Experiments show that the data augmentation method can significantly improve the performance of the pre-trained models on the multilingual multimodal reasoning task.

3. Experiments and details are rich.

**Reasons To Reject:**

1. Compared with the baseline， the performance degradation of the ablation models is not obvious, and it is unknown which data augmentation strategy plays a role in performance improvement.

2. This work proposed a concept-based mixup method to expand image data, which identifies visual elements that correspond to high-resource concepts and replaces them with visual elements associated with the original low-resource concept. However, I did not find the relevant details in the paper.

**Reproducibility:**

3: Could reproduce the results with some difficulty. The settings of parameters are underspecified or subjectively determined; the training/evaluation data are not widely available.

**Reviewer Confidence:**

4: Quite sure. I tried to check the important points carefully. It's unlikely, though conceivable, that I missed something that should affect my ratings.

---

> ### Author Rebuttal · Authors · 2023-08-29
>
> Dear Reviewer bqm3,
>
> Thank you for your thoughtful and thorough review of our manuscript. We highly appreciate the time and effort you put into evaluating our work, and your comments are instrumental in improving the paper. Below, we respond to each point of concern you raised.
>
> **Question1**: Compared with the baseline method mUNITER, the performance degradation of all ablation models was not obvious. Which data augmentation strategy played a role in performance improvement? Might it be possible to ablate multiple data augmentation strategies simultaneously to observe changes in model performance?
>
> **Reply**:The impression that the degradation in performance for our ablation models is not significant compared to the mUNITER baseline deserves clarification.
>
> 1. **Degradation is More Significant than It Appears**:  While it may appear that the decline in performance is subtle (about 1-2 points), it's crucial to consider this within the context of the overall performance gain, which is less than 4 points. Given this, a 1-2 point drop is indeed substantial. As depicted in Table 5 of our paper, the impact of omitting phrase and image information is substantial. For example, when we removed concept and phrase information, the performance showed a more significant decrease compared to other ablations.
> 2. **Multiple Ablations Conducted**: Taking your suggestion to heart, we conducted additional experiments to ablate multiple data augmentation strategies simultaneously. The table provided showcases these results.
>
> | Method | NLVR2 | MaRVL (ZH, TA, SW, ID, TR, avg.) |
> | --- | --- | --- |
> | Baseline | 70.93 | 54.05, 53.14, 51.72, 55.41, 53.22, 53.51 |
> | Our Method | 71.56 | 59.71, 58.45, 54.35, 55.24, 58.81, **57.31** |
> | All w/o Concept | 71.18 | 58.87, 57.30, 53.82, 54.61, 57.52, **56.42** |
> | All w/o Phrase | 70.71 | 57.95, 56.94, 53.18, 54.02, 57.19, 55.86 |
> | All w/o Word | 70.65 | 58.69, 57.22, 53.55, 54.38, 57.55, 56.28 |
> | All w/o Image | 70.85 | 58.14, 56.81, 52.95, 54.69, 56.33, 55.78 |
> | All w/o Concept+Phrase | 69.91 | 56.27, 54.92, 53.05, 53.78, 55.63, 54.73 |
> | All w/o Concept+Word | 71.08 | 57.91, 57.03, 53.44, 53.72, 54.41, 55.30 |
> | All w/o Concept+Image | 70.39 | 57.21, 55.99, 52.75, 54.36, 55.07, 55.07 |
> | All w/o Phrase+Word | 70.94 | 57.81, 55.22, 53.37, 53.89, 56.51, **55.36** |
> | All w/o Phrase+Image | 70.85 | 57.25, 55.77, 52.31, 53.11, 55.48, 54.78 |
> | All w/o Image+Word | 69.88 | 57.94, 55.35, 52.41, 53.49, 56.27, 55.09 |
> | All w/o Concept+Phrase+Word | 71.10 | 56.01, 54.38, 52.88, 53.56, 54.35, 54.24 |
> | All w/o Concept+Word+Image | 70.68 | 56.95, 55.01, 52.51, 53.36, 54.27, **54.42** |
> | All w/o Concept+Image+Phrase | 70.72 | 56.12, 54.36, 51.91, 53.07, 53.83, 53.86 |
> | All w/o Phrase+Word+Image | 70.45 | 55.97, 54.81, 52.05, 53.16, 54.93, 54.18 |
>
>
>
> **Key Insights from Multiple Ablations:**
>
> 1. **Two-Feature Ablations**:
>     - Removing both 'Concept' and 'Phrase' has the most significant impact on performance.
>     - Conversely, the least effect is seen when 'Phrase' and 'Word' are removed, indicating that pairs of cultural concepts and their corresponding images are most beneficial for performance improvement.
> 2. **Three-Feature Ablations**:
>     - Ablating 'Concept', 'Image', and 'Phrase' together has the largest impact, suggesting that the 'Word' feature contributes least to the performance gains.
>     - Removing 'Concept', 'Word', and 'Image' has the least impact, signifying that the 'Phrase' feature contributes most to performance gains.
>
> These multiple ablations provide nuanced insights into the importance of each feature in our model. They allow us to isolate and understand how individual or combinations of features contribute to performance. We hope this detailed breakdown addresses your question effectively.
>
> **Question2**: I did not find relevant details about the Concept-Based Mixup method in the paper. What object detection algorithm do you use to identify images? How do you replace identified high-resource image elements with low-resource cultural image elements?
>
> **Reply**: Thank you for your question regarding the Concept-Based Mixup method, a topic that, unfortunately, was omitted in our original paper due to space constraints. Your inquiry has made us realize the importance of including this information, and we apologize for the oversight. Rest assured, this section will be incorporated into the revised version of the paper.
>
> To address your question:
>
> 1. **Object Detection Model**: We use the Detic[1] model available at [Facebook Research's GitHub repository](https://github.com/facebookresearch/Detic) for object detection. It employs the CLIP[2] classifier and offers open-vocabulary capabilities. [Here is their official demo using Colab for further reference.](https://colab.research.google.com/drive/1QtTW9-ukX2HKZGvt0QvVGqjuqEykoZKI)
> 2. **Vocabulary Configuration**: With this model, we can freely configure the vocabulary, allowing us to detect specific cultural concepts within images.
>
> For the actual mixup process, here's our approach:
>
> 1. **Visual Element Detection**: We use the high-resource concept as the vocabulary term and deploy the object detection model to locate the corresponding visual elements in the image.
> 2. **Bounding Box Extraction**: Based on the object detection results, we extract the object that needs to be replaced from the original high-resource image using bounding boxes.
> 3. **Resizing**: We adjust the size of the low-resource image element, which is collected together with concepts themselves, to match the size of the bounding box.
> 4. **Pasting**: The resized new low-resource image is then pasted into the corresponding location of the original image.
> 5. **Edge Smoothing**: Finally, Poisson blending is applied to smoothly integrate the new object with the original image.
>
> We hope this clarifies how the Concept-Based Mixup method is implemented in our experiments. Thank you for bringing attention to this crucial aspect of our work.
>
> [1]Detecting Twenty-thousand Classes using Image-level Supervision
>
> [2]Learning Transferable Visual Models From Natural Language Supervision
>
>  **Reproducibility**
> We have taken note of your valuable feedback concerning the reproducibility of our study. Should our paper be accepted, we will release both the code and the data to aid the scientific community in replicating our results. We believe this will address your concerns regarding the replicability of the study. Thank you for bringing this important issue to our attention.
>
> In conclusion, we truly value your constructive feedback, which has guided substantial improvements in our manuscript. If our responses have been satisfactory in resolving your questions, we kindly ask that you consider re-evaluating our paper. Specifically, we would be extremely grateful if you could increase your scoring for 'Soundness' and 'Excitement.' We look forward to any further comments you may have.
>
> Sincerely,
>
> All Authors of the Paper

---

### Official Review · Reviewer_sDGb · 2023-08-04

**Typos Grammar Style And Presentation Improvements:** N/A
**Soundness:** 4

**Excitement:**

4: Strong: This paper deepens the understanding of some phenomenon or lowers the barriers to an existing research direction.

**Missing References:**

N/A

**Paper Topic And Main Contributions:**

This paper aims to alleviate the problem of unsatisfactory multicultural competence of low-resource languages in multimodal reasoning. To this end, it proposes Cultural Concept Adaptation with two steps: firstly, adapting an annotation-free method for cultural-concept adaptation and constructing a concept mapping set; secondly, utilizing the three-tier code-switching strategy and cultural concept-based mix-up for the purpose of multimodal data augmentation. The proposed method achieved improved performance in zero-shot and few-shot settings on several low-resource languages (relative to English).

**Questions For The Authors:**

* The possibility and portion of components in the text being replaced by Three-tier Code-switching can be stated in more detail. It will be even better when there are hyper-parameter experiments about the possibility or the portion of the replaced components gradually changing on a scale.
* As LLMs are generally good when doing something like culture concepts explanation. E.g., they can answer what is an erhu. Why not adapt ChatGPT to do the cultural concept adaption? If ChatGPT is not a good expert, it can also be a baseline for the scenario.

**Reasons To Accept:**

* Proposed to enhance the multicultural realization and inability of models to alleviate the undesired multicultural performance due to the absence of other non-English languages' sufficient data.
* Proposed a multimodal culture-related augmentation method with two steps to solve the observed problem.
* Promising performance on the target multimodal reasoning task with reasonable experiments.
* Good writing.

**Reasons To Reject:**

* More details and even more experiments are required to explore the boundary of the proposed method.

**Reproducibility:**

4: Could mostly reproduce the results, but there may be some variation because of sample variance or minor variations in their interpretation of the protocol or method.

**Reviewer Confidence:**

3: Pretty sure, but there's a chance I missed something. Although I have a good feel for this area in general, I did not carefully check the paper's details, e.g., the math, experimental design, or novelty.

---

> ### Author Rebuttal · Authors · 2023-08-29
>
> Dear Reviewer sDGb,
>
> Thank you for your thorough review and feedback on our manuscript. We truly appreciate the time and effort you dedicated to understanding our work, and your constructive comments. We'd like to address each of your concerns:
>
> **Question 1**: The possibility and portion of components in the text being replaced by Three-tier Code-switching can be stated in more detail. It will be even better when there are hyper-parameter experiments about the possibility or the portion of the replaced components gradually changing on a scale.
>
> **Reply** : Thank you for your question regarding the need for more detailed information on the possibility and portion of text components replaced by the Three-tier Code-switching technique. You've raised a valid point, and I appreciate the opportunity to provide further clarification.
>
> **Regarding Possibility:**
> We have collected detailed statistics from multiple angles to address your query. In our dataset tailored for low-resource cultural adaptation:
>
> *Concept*: The probability that a cultural concept from low-resource cultural adaptation set appears at least once in the training set is 64.3%. The sentences that could potentially be retrieved to implement this concept constitute around 75% of the entire training dataset.
>
> *Phrase*: Among these retrievable sentences, at least one phrase that can be replaced exists with a probability of 78.5%.
>
> *Word*:  In terms of at least one  words that can be replaced, the probability stands at 97.8%. On average, about 70% of the words in a sentence can find possible low-resource replacements.
>
> **Regarding Portion:**
> In our paper, we have formalized the methodology in such a way that it is adaptable to a broad range of scenarios, making use of probability distributions for a more nuanced representation of the methods involved. However, in the experiments, specific data requires specific analysis. For instance, in the context of our dataset:
> 1. In terms of the dataset composition, we follow a 15:5 ratio. That means 75% of the data in the training set is in English, and the remaining should be as diverse as possible.
> 2. For retrievable sentences with cultural concepts and phrases, usually only one to two phrases per sentence can be replaced. Also, only 70% of the words can be queried for possible replacements, thereby setting a natural upper limit.
>
> Given these, we set the replacement probability for Three-tier Code-switching at 100%. This not only yielded the best results but also eliminated the need for complicated hyperparameter tuning.
>
> **Hyperparameter Experiments:**
>
> As per your suggestion, we also experimented with different probabilities for the replacement and observed consistent performance improvements across all three models as the sampling probability increased:
>
> | Sampling Probability | mUNITER | UC2 | M3P |
> | --- | --- | --- | --- |
> | 0% | 54.0 | 57.28 | 56.0 |
> | 30% | 55.32 | 58.14 | 57.46 |
> | 50% | 56.27 | 58.85 | 58.79 |
> | 70% | 57.04 | 59.16 | 59.34 |
> | 100% | 57.31 | 59.33 | 59.79 |
>
> Also, modifying the sampling ratios at individual levels—such as phrases, concepts, or words—does have an impact on the final performance, as clearly evidenced by our ablation studies.
>
> In summary, we find that higher probabilities for replacement are beneficial for the model performance. We hope this addresses your question comprehensively. Thank you for the insightful inquiry, which has provided an opportunity for us to clarify our methodology.
>
> **Question 2**:  As LLMs are generally good when doing something like culture concepts explanation. E.g., they can answer what is an erhu. Why not adapt ChatGPT to do the cultural concept adaption? If ChatGPT is not a good expert, it can also be a baseline for the scenario.
>
> **Reply** : Thank you for the thoughtful question regarding the potential of using large language models (LLMs) like ChatGPT for cultural concept adaptation. While it's true that LLMs can perform well in explaining various cultural concepts, there are several considerations that guided our decision to not solely rely on them as a baseline or as an expert system. Allow me to elaborate:
>
> 1. **Language Resource Limitations**: Previous studies [1][2][3] and our own  case studies on the GPT-3.5 of initial version have shown that the model performs sub-optimally on some low-resource languages. For example, it may perform well for languages like Chinese (zh), it shows limitations for languages like Swahili (sw).
> 2. **Cost-Effectiveness**: Dictionary lookup operations are generally more efficient and cheaper compared to running large language models.
> 3. **Reproducibility & Hallucinations Issues**: LLMs have inherent weaknesses. They are not very stable based on prompts, making it difficult to guarantee reproducibility. Additionally, verifying the model's outputs can be challenging, particularly when hallucinations occur.
> 4. **Dataset Compatibility**: Our downstream dataset, NLVR2, is constructed around a subset of WordNet, making it easier to query. If we were to rely on ChatGPT, additional processing would be needed to match the output.
> 5. **Leveraging Existing Dictionary Resources**: Our method have the potential to make full use of existing dictionary resources for extremely low-resource languages, such as Navajo, through platforms like ConceptNet.
>
> Beyond these concerns, recently, we have evaluated the latest GPT-3.5 and GPT-4 models for their capability in cultural adaptation. We employed a chain of prompt approach to construct a cultural adaptation graph and found promising results, especially with GPT-4. However, these models also face the challenge of potential hallucinations and require careful prompt design to generate diverse cultural adaptations.
>
> In conclusion, we believe that a more reasonable approach would be to explore the combination of dictionary-based cultural adaptation methods with large language models. For example, the dictionary-based method could generate a large set of cultural adaptations, which could then be filtered for relevance by the large language model. Incorporating large language models to construct our proposed cultural adaptation graph does indeed hold promise and is a direction we are considering for future work.
>
> [1] Is ChatGPT A Good Translator? A Preliminary Study
>
> [2] Prompting PaLM for Translation: Assessing Strategies and Performance
>
> [3] Assessing Cross-Cultural Alignment between ChatGPT and Human Societies: An Empirical Study
>
> [4] Multilingual Language Models are not Multicultural: A Case Study in Emotion
>
> **Reproducibility Concerns:**
>
> We understand the importance of reproducibility in research. To this end, we'll work on providing a more detailed specification of our experimental setup, including a clearer delineation of parameter settings and the rationale behind them. And we will release both the code and the data to aid the scientific community in replicating our results.
>
>
> Once again, we are grateful for your constructive feedback and insights. They provide us with a clear direction on how to refine and enhance our paper. We are optimistic that the revisions which integrated your valuable suggestions will further elevate the impact of our work.
>
> Warm regards,
>
> All Authors of the Paper

---

### Official Review · Reviewer_eiJt · 2023-08-07

**Soundness:** 3

**Excitement:**

4: Strong: This paper deepens the understanding of some phenomenon or lowers the barriers to an existing research direction.

**Missing References:**

Sun, J., Ahn, H., Park, C. Y., Tsvetkov, Y., & Mortensen, D. R. (2021, April). Cross-Cultural Similarity Features for Cross-Lingual Transfer Learning of Pragmatically Motivated Tasks. In Proceedings of the 16th Conference of the European Chapter of the Association for Computational Linguistics: Main Volume (pp. 2403-2414).

**Paper Topic And Main Contributions:**

This paper focuses on multimodal reasoning over images and languages, with an emphasis on low-resource cultural concepts. Specifically, the author(s) introduce a cultural-concept adaptation method to align low-resource cultural concepts with high-resource ones based on their shared semantic properties. With the resulting concept mapping set, the author(s) further propose a multimodal data augmentation method based on the code-switching strategy. Experiments on a popularly used multicultural multimodal reasoning benchmark (MaRVL) show that the proposed method improves the baselines in both zero- and few-shot settings.

**Questions For The Authors:**

Question A: Although the author(s) provide a description of cultural concept collection in Appendix A, the description primarily focuses on the data resources and considered geo-cultures as well as cultural categories. I'm still confused about how the author(s) selected low-resource cultural concepts from the mentioned resources. E.g., manually or automatically?How to guarantee the collected concepts' quality?

Question B:  Regarding cultural-level code-switching for data augmentation, the author(s) only considered the replacement of high-resource cultural concepts by low-resouce ones. In fact, this strategy can further extend to any two low-resource cultures (e.g., 'Erhu' in China versus 'Kongahyan' in Indonesian), possibly augmenting more data and benefiting the model with a deep understanding of low-resource culural concepts. I'm wondering if there are some potential reasons for author(s) emphasize high-resource versus low-resource cultural concept code-switching?

Question C: Although the authors show the benefits of cultural-adapted data augmentation on baselines for multimodal reasoning, the quality of the generated cultural adaptation graph is uncertain. Did the author(s) conduct any evaluation (e.g., human evaluation) for this generated graph quality?

Question D: Did the author(s) conduct any error analyis to explore potential patterns of model prediction errors?

**Reasons To Accept:**

1. The idea of capturing concept pairs for cultural adaptation is interesting.

2. The author(s) conducted a bunch of experiments to evaluate and analyze the proposed method.

3. The paper is easy to follow.

**Reasons To Reject:**

Some fundamental details in the method and experiment design are missing. Please see questions in the next section.

**Reproducibility:**

4: Could mostly reproduce the results, but there may be some variation because of sample variance or minor variations in their interpretation of the protocol or method.

**Reviewer Confidence:**

4: Quite sure. I tried to check the important points carefully. It's unlikely, though conceivable, that I missed something that should affect my ratings.

---

> ### Author Rebuttal · Authors · 2023-08-29
>
> Dear Reviewer eiJt,
>
> Thank you for taking the time to review our paper and we appreciate the positive feedback and your detailed questions regarding our methodology and experiments. We would like to address your concerns as follows:
>
> **Question A**: Although the author(s) provide a description of cultural concept collection in Appendix A, the description primarily focuses on the data resources and considered geo-cultures as well as cultural categories. I'm still confused about how the author(s) selected low-resource cultural concepts from the mentioned resources. E.g., manually or automatically?How to guarantee the collected concepts' quality?
>
> **Reply**: Thank you for your question regarding the method of collecting cultural concepts outlined in Appendix A. We understand your concern about how the concepts are selected, especially in the context of low-resource cultural information, and the assurance of quality.
>
> To clarify, the collection of cultural concepts is primarily carried out manually. Importantly, this approach doesn't necessitate cross-cultural specialists; rather, it employs crowd-sourced workers familiar with the respective culture, making it a relatively cost-effective and simple process. The procedure is twofold:
>
> 1. **Diverse Candidate Collection**: For each culture under consideration, we involve five crowd-sourced workers who follow the guidelines specified in the appendix A. They collect as many cultural concepts as possible for each category. The collected data from each worker is then aggregated.
> 2. **Voting for Filtering**: Next, an additional 10 crowd-sourced workers from the respective country or region assess whether each gathered concept genuinely belongs to the local culture. If seven or more evaluators agree, the concept is added to the final 'culture concept set.'
>
> This method ensures quality in several ways:
>
> - **Inclusion of Multiple Evaluators**: By involving multiple people from the respective culture in both the collection and the validation process, we minimize individual bias and enhance the dataset's reliability.
> - **Threshold for Inclusion**: The use of a voting system provides a safety net against inaccuracies and biases. If a concept is included, it's because a significant majority (at least 7 out of 10) of the evaluators from that culture have vouched for its relevance.
> - **Manual Over Automatic**: While automated methods may miss nuances or make errors, manual collection engages those who understand the cultural intricacies best—the people from that culture.
>
> Hence, our approach offers a robust, yet economical way of collecting high-quality cultural concepts. We hope this clears up any confusion you may have had.
>
> **Question B**: Regarding cultural-level code-switching for data augmentation, the author(s) only considered the replacement of high-resource cultural concepts by low-resouce ones. In fact, this strategy can further extend to any two low-resource cultures (e.g., 'Erhu' in China versus 'Kongahyan' in Indonesian), possibly augmenting more data and benefiting the model with a deep understanding of low-resource culural concepts. I'm wondering if there are some potential reasons for author(s) emphasize high-resource versus low-resource cultural concept code-switching?
>
> **Reply**: Your insight about extending the data augmentation strategy to low-resource versus low-resource cultural concept code-switching is both perceptive and innovative. However, there are several underlying reasons why we emphasized high-resource versus low-resource cultural concept code-switching.
>
> 1. **High-Resource as a "Bridge"**: For most open-source multilingual and multimodal models, a significant proportion of the pre-training data is in English, which naturally incorporates a wealth of cultural knowledge. Using this high-resource English data as a "bridge" is logical for aiding the adaptation of low-resource cultural concepts.
> 2. **Downstream Task Constraints**: Many downstream task datasets used for evaluation have limited representation of low-resource cultures. For instance, the NLVR2 dataset is solely in English, making quantitative evaluation between low-resource cultures difficult.
>
> To address your constructive suggestion:
>
> 1. **Theoretical Feasibility**: Theoretically, the idea holds water. The term "low-resource" is relative, and code-switching between different low-resource cultures could potentially benefit each other.
> 2. **Proof of Concept**: In line with your suggestion, we translated the NLVR2 dataset into Chinese via machine translation and applied the Cultural Adaptation method (as detailed in section 3.1 of the paper). We then created cultural adaptation sets from Indonesian (ID), Swahili (SW), Tamil (TA), and Turkish (TR) to Chinese. Following this, we employed the Concept-Based Data Augmentation from section 3.2, using a 15:5 ratio for augmentation and the model is mUNITER. Compared to a baseline model trained solely on the translated Chinese training set, our method exhibited improved performance on the MARVL test across all the involved languages:
>     - Chinese (ZH): Baseline 62.61, Our method 61.88
>     - Indonesian (ID): Baseline 53.20, Our method 56.17
>     - Swahili (SW): Baseline 51.42, Our method 53.34
>     - Tamil (TA): Baseline 52.11, Our method 57.58
>     - Turkish (TR): Baseline 54.26, Our method 57.93
>
> Thus, our proof-of-concept trial supports the merit in extending the data augmentation strategy beyond just high-resource to low-resource cultural concept substitution. We hope this clarifies why we initially emphasized high-resource vs. low-resource concepts while also acknowledging the validity and potential of your suggestion.
>
> **Question C**: Although the authors show the benefits of cultural-adapted data augmentation on baselines for multimodal reasoning, the quality of the generated cultural adaptation graph is uncertain. Did the author(s) conduct any evaluation (e.g., human evaluation) for this generated graph quality?
>
> **Reply**: You raise an important question regarding the quality of the generated cultural adaptation graph, a key component of our methodology. Indeed, we share your concern for ensuring high-quality output in this aspect of our work.
>
> Before proceeding with data augmentation experiments, we did conduct a human evaluation of the generated cultural adaptation graphs. The results were quite reassuring:
>
> 1. **High Human Agreement**: Approximately 84% of the generated cultural adaptation graphs were in alignment with human judgment. This high rate of agreement underscores the validity of our approach in capturing culturally meaningful concepts.
> 2. **Addressing Inaccuracies**: For the remaining graphs that were less accurate, the primary issue often lay in the lack of direct English translations for the low-resource cultural concepts' hypernyms. To address this, the method would need to consider higher-order hypernyms, making the entire path too distant and potentially distorting cultural similarity according to human evaluators.
>
> To counteract these issues:
>
> - **Hypernym Limitation**: As described in the paper, we restrict the method to considering at most three-order hypernyms during the construction of the cultural adaptation graph. If a suitable translation or hypernym cannot be identified within this constraint, the concept is discarded.
> - **Exponential Decay of Sampling Probability**: The paper also outlines that the sampling probability for including concepts in the cultural adaptation set decays exponentially with path distance. This mechanism serves to mitigate possible inaccuracies by giving greater weight to more closely related concepts.
>
> These measures are designed to ensure that the cultural adaptation graph generated is of high quality, both in terms of capturing authentic cultural elements and in conforming to human judgment. Therefore, while the graph is not perfect, it is constructed with numerous safeguards to ensure its utility and accuracy. I hope this clarifies any concerns you might have had.
>
> **Question D**: Did the author(s) conduct any error analyis to explore potential patterns of model prediction errors?
>
> **Reply**: You make a valid point about the importance of error analysis to understand the model's limitations. While we did conduct an error case analysis, we had to omit it from the paper due to space constraints. However, we are eager to share some of the key findings here to address your concerns.
>
> 1. **Performance Across Languages**: As Table 3 in the paper shows, the performance varies across languages. This is influenced by the imbalanced distribution of the pre-training data as well as linguistic and cultural differences. The ease of transfer from English to other languages is also non-uniform.
> 2. **Within-Language Category Differences**: When looking within a single language, we found that the difficulty varies across different categories. For instance, the model performs better on "animal and plant" and "building" categories compared to others. As detailed in section 4.3.4 of the paper ("Reduce Model Bias"), this also relates to the distribution of pre-training data.
> 3. **Instance-Level Analysis**: We conducted a more granular error analysis at the instance level and found insightful patterns:
>     - **Poor Spatial Awareness**: The model generally struggles with spatial reasoning. Specifically, if the description format is "one image..., another image..." instead of specifying "left image..., right image...", the models statistically perform better across languages (58.4%  vs 53.8% ).
>     - **Limited Textual Reasoning**: The model also fares better when the description is given in a single sentence ("in both images...") as opposed to two separate ones ("one image..., another image..."). Here, the model's performance increased to 60.4%.
>     - **False Bias**: The models have a higher tendency to predict 'false' in the test set. Specifically, 57% of the model predictions were 'false' and 43% were 'true', even though the actual distribution of 'true' and 'false' labels in both the training and test sets is balanced at 50%. The main reason for this skew is the model's unfamiliarity with specific cultural concepts and images, leading it to predict 'false'.
>
> These analyses offer a multi-level perspective into the shortcomings of the model and help us understand where improvements can be made. We hope this clarifies any concerns you may have had regarding the model's error characteristics.
>
> **Missing References:**
>
> Thank you for pointing out the missing reference. We are aware of Sun et al.'s work and, upon reflection, agree that it should be included in our discussions. We will make sure to incorporate this in our revised version.
>
> **Looking Forward to Your Reply and Further Discussion**
>
> We sincerely hope that our clarifications adequately address your concerns. As we move to revise our manuscript, we will integrate your valuable suggestions to further strengthen the research presented. If our responses have been satisfactory in resolving your questions, we kindly ask that you consider re-evaluating our paper. Specifically, we would be extremely grateful if you could revisit your scoring for 'Soundness' and 'Excitement.'
>
> Should you have any further concerns or require additional clarifications, we are more than willing to engage in further discussions to ensure the quality of our research.
>
> Warm regards,
>
> All Authors of the Paper

---

### Meta-Review · Area_Chair_EseS · 2023-09-19

**Recommendation:** 4

**Metareview:**

(I've used bqm3's updated soundness score of 4, which was mentioned in the comments but not reflected in the numeric score).

The paper focuses on the problem of cultural adaptation in multimodal reasoning, in which some concepts are difficult to translate directly from one language to another. They suggest an annotation-free process for augmenting data using a "code switching" approach. The reviewers are unanimous that the experiments provided are well done and provide support for the authors' claims about the benefits of their approach. The detailed author responses clarify outstanding questions and suggest careful thinking and thorough work.

The reviewers raised several significant questions about the details of the work, including the source of the cultural concepts they use. The authors provide more details about the crowdsourced data collection effort, which is also described in Appendix A.

Several other questions that reviewers raised are thoroughly answered by the authors. Due to space constraints, not all of the answers may appear in the final text, but seem sufficient to address the reviewer questions.

---

### Decision · Program_Chairs · 2023-10-07

**Decision:**

Accept-Main

**Comment:**

(I've used bqm3's updated soundness score of 4, which was mentioned in the comments but not reflected in the numeric score).

The paper focuses on the problem of cultural adaptation in multimodal reasoning, in which some concepts are difficult to translate directly from one language to another. They suggest an annotation-free process for augmenting data using a "code switching" approach. The reviewers are unanimous that the experiments provided are well done and provide support for the authors' claims about the benefits of their approach. The detailed author responses clarify outstanding questions and suggest careful thinking and thorough work.

The reviewers raised several significant questions about the details of the work, including the source of the cultural concepts they use. The authors provide more details about the crowdsourced data collection effort, which is also described in Appendix A.

Several other questions that reviewers raised are thoroughly answered by the authors. Due to space constraints, not all of the answers may appear in the final text, but seem sufficient to address the reviewer questions.